# Continuous Glucose Monitoring in the Intensive Care Unit Following Total Pancreatectomy with Islet Autotransplantation in Children: Establishing Accuracy of the Dexcom G6 Model

**DOI:** 10.3390/jcm10091893

**Published:** 2021-04-27

**Authors:** Natalie Segev, Lindsey N. Hornung, Siobhan E. Tellez, Joshua D. Courter, Sarah A. Lawson, Jaimie D. Nathan, Maisam Abu-El-Haija, Deborah A. Elder

**Affiliations:** 1Cincinnati Children’s Hospital Medical Center, Cincinnati, OH 45229, USA; 2Division of Biostatistics and Epidemiology, Cincinnati Children’s Hospital Medical Center, Cincinnati, OH 45229, USA; Lindsey.Hornung@cchmc.org; 3Division of Endocrinology, Department of Pediatrics, Cincinnati Children’s Hospital Medical Center, Cincinnati, OH 45229, USA; Siobhan.Tellez@cchmc.org (S.E.T.); Sarah.Lawson@cchmc.org (S.A.L.); Deborah.Elder@cchmc.org (D.A.E.); 4Division of Pharmacy, Cincinnati Children’s Hospital Medical Center, Cincinnati, OH 45229, USA; Joshua.Courter@cchmc.org; 5Division of Pediatric General and Thoracic Surgery, Cincinnati Children’s Hospital Medical Center, Cincinnati, OH 45229, USA; Jaimie.Nathan@cchmc.org; 6Department of Surgery, College of Medicine, University of Cincinnati, Cincinnati, OH 45229, USA; 7Division of Pediatric Gastroenterology, Cincinnati Children’s Hospital Medical Center, Cincinnati, OH 45229, USA; Maisam.Haija@cchmc.org; 8Department of Pediatrics, College of Medicine, University of Cincinnati, Cincinnati, OH 45229, USA

**Keywords:** continuous glucose monitoring, total pancreatectomy with islet autotransplantation, mean absolute relative difference, mean absolute difference, Dexcom G6

## Abstract

Hyperglycemia is detrimental to postoperative islet cell survival in patients undergoing total pancreatectomy with islet autotransplantation (TPIAT). This makes continuous glucose monitoring (CGM) a useful management tool. We evaluated the accuracy of the Dexcom G6 CGM in pediatric intensive care unit patients following TPIAT. Twenty-five patients who underwent TPIAT had Dexcom G6 glucose values compared to paired serum glucose values. All paired glucose samples were obtained within 5 minutes of each other during the first seven days post TPIAT. Data were evaluated using mean absolute difference (MAD), mean absolute relative difference (MARD), %20/20, %15/15 accuracy, and Clarke Error Grid analysis. Exclusions included analysis during the CGM “warm-up” period and hydroxyurea administration (known drug interference). A total of 183 time-matched samples were reviewed during postoperative days 2–7. MAD was 14.7 mg/dL and MARD was 13.4%, with values of 15.2%, 14.0%, 12.1%, 11.4%, 13.2% and 14.1% at days 2, 3, 4, 5, 6 and 7, respectively. Dexcom G6 had a %20/20 accuracy of 78%, and a %15/15 accuracy of 64%. Clarke Error Grid analysis showed that 77% of time-matched values were clinically accurate, and 100% were clinically acceptable. The Dexcom G6 CGM may be an accurate tool producing clinically acceptable values to make reliable clinical decisions in the immediate post-TPIAT period.

## 1. Introduction

Continuous glucose monitoring (CGM) systems provide comprehensive blood glucose information to individuals with diabetes mellitus [1]. Advances in continuous glucose monitor technology are focused on improving accuracy with the goal of reducing, or replacing, fingerstick testing by glucometer [2]. The CGM system consists of a sensor, transmitter, and receiver. The sensor device adheres to the skin and secures a small, flexible sensor in place beneath the skin. Using electrode signals through interstitial fluid, the sensor reports a glucose value every 5 minutes to a receiving device (or smart device) via the attached transmitter, providing the wearer with real-time glucose values and data trends. This system allows for frequent sampling of calculated glucose levels without the need for repetitive fingerstick testing by glucometer. The Dexcom G6 CGM system (Dexcom Inc., San Diego, CA, USA) is one of two FDA-approved systems to replace fingerstick glucose monitoring without requiring the entry of glucometer values for calibration [3,4,5,6,7,8,9,10]. Both the FreeStyle Libre and Dexcom G6 system have similar reported accuracies, however the FreeStyle Libre was not approved for use in children at the initiation of this study [3,4].

Recent improvements to the Dexcom G6 CGM include indications for children who are at least 2 years of age, improved accuracy with longer wearer use (up to 10 days), and improved sensor membrane with limited interference from acetaminophen co-administration. Dexcom reports improvement in accuracy compared to prior models by citing a mean absolute relative difference (MARD) of 9.0% for the G6 system [7]. The MARD has become the measure of choice for evaluating the accuracy of various glucose measurement devices by comparing an average of the absolute differences to a gold standard time-matched collected value (i.e., serum glucose values). In addition to MARD, Dexcom reports accuracy by evaluating the proportion of CGM glucose values that are within 20% of time-matched reference values when >100 mg/dL or within 20 mg/dL of time-matched reference values when ≤100 mg/dL (referred to as %20/20) [3,7].

Since CGM technology is generally used in the outpatient setting, further studies are needed to explore its accuracy in critically ill patients and various disease states. In this study, we explore the use of the Dexcom G6 CGM technology in patients who have undergone total pancreatectomy with islet cell autotransplantation (TPIAT), a post-operative population that requires careful glucose management. TPIAT is offered to patients whose quality of life has been significantly impacted by acute recurrent pancreatitis or chronic pancreatitis, which in children is often caused by a genetic mutation [11,12,13]. Reports document that TPIAT results in alleviation of pain, decreased opioid dependence, and improved quality of life in these patients [14,15]. The islet autotransplantation allows for the potential to achieve insulin independence and decreases the risk of brittle diabetes long term [11,14,15,16,17,18,19]. During the islet autotransplantation, islets are isolated from the pancreas, infused into the portal vein, and allowed to engraft in the liver [20]. While awaiting engraftment, patients require exogenous insulin therapy for strict glycemic control. Because extreme fluctuations of hyperglycemic environments are detrimental to the survival of transplanted islets [21], CGM technology can be a useful tool to aid in management post-TPIAT. There have been studies of other CGM models in their use post-TPIAT that demonstrate accuracy [5,6]. In previous work, we reported a study of the Dexcom G4 model in patients after TPIAT in the immediate postoperative period in the pediatric intensive care unit (PICU), reporting MARD of 10.6% [6]. There has yet to be data in the TPIAT population on the accuracy of the newest Dexcom G6 model. In this study, our primary objective is to establish accuracy of the Dexcom G6 in the immediate postoperative period after TPIAT in the PICU setting.

## 2. Materials and Methods

### 2.1. Study Design and Procedures

A retrospective cohort chart review was performed to collect data from the medical records of all patients who underwent TPIAT from October 2019 to February 2021 at Cincinnati Children’s Hospital Medical Center (CCHMC). During this time period, 25 patients underwent TPIAT, and data from the first seven postoperative days in the PICU were collected. CCHMC IRB approval was obtained for this study (2019-0608). Inclusion criteria for this study encompassed all patients admitted to the CCHMC PICU post-TPIAT. All serum glucose and CGM values collected within 5-minute intervals in the first seven days post-op were included in data collection and analysis. Exclusion criteria encompassed a single patient who was on dialysis at the time of surgery and during the post-operative course, and any blood glucose and CGM values that were collected after initiation of hydroxyurea medication.

Sensor readings from a Dexcom G6 CGM system were compared to paired serum glucose values. During the TPIAT operation, patients were started on an intravenous insulin infusion, and this was continued postoperatively. Insulin was titrated based on hourly-obtained glucose values (either point-of-care (POC) fingerstick glucoses or obtained serum glucoses) to maintain glycemic control within the desired acceptable blood glucose range (70–140 mg/dL). Insulin was not titrated based on CGM values alone as the accuracy of the Dexcom G6 in our post-operative TPIAT population had not yet been studied. A Dexcom G6 CGM was placed on the anterior thigh of each patient once they arrived in the PICU and was used to monitor glucose trends. The abdomen was not used as a site of Dexcom placement, to avoid proximity to the site of the surgical incision which would interfere with appropriate post-operative wound care and avoid insertion into abdominal wall edema which could potentially affect CGM readings. Arms were avoided due to vascular access and blood pressure monitoring. Serum glucose was measured and matched to the closest CGM value within the immediate five minutes before or after serum glucose was obtained in order to compare time-matched measures. CGM values were excluded from this study after the initiation of hydroxyurea due to the influence the medication can have on the G6 sensor [22,23]. Primary outcomes of this study included comparison of CGM values to serum glucose values and therefore determining Dexcom G6 CGM accuracy (through the use of MARD, %20/20, and %15/15). Secondary outcomes include determining whether the Dexcom G6 CGM is a clinically acceptable tool to use for clinical decision-making (through the use of Clarke Error Grid analysis).

### 2.2. Statistical Analysis

Data were analyzed using SAS^®^, version 9.4 (SAS Institute, Cary, NC, USA). Continuous data were summarized as means with standard deviations (SD) or medians with interquartile ranges (IQR: 25th–75th percentiles), while categorical data were summarized as frequency counts and percentages. Both the mean absolute difference (MAD) and the MARD were calculated to compare CGM values to time-matched serum glucose values. Data were divided into CGM values that were within ±20% of time-matched serum glucose value if >100 mg/dL and ±20 mg/dL of time-matched serum glucose if ≤100 mg/dL (referred to as %20/20), in addition to a corresponding %15/15. Dexcom G6 manufacturer cautions that day 1 use functions as a “warm-up” period, and accuracy is greatly improved beginning on day 2 [24]. Data from postoperative days 1–7 were analyzed and data from postoperative days 2–7 were separately analyzed due to known increased variability during CGM day 1. This seven day time period reflects the average time before hydroxyurea was prescribed to lower serum platelet counts due to known CGM-hydroxyurea interference [23].

Glucose values were considered to be in goal range if they were between 70 and 140 mg/dL and were considered to be out-of-range if they were <70 mg/dL or >140 mg/dL. Clarke Error Grid analyses were performed to compare CGM values to serum glucose values. Zone A is considered to be clinically accurate and leads to clinically correct treatment decisions. Zone B, though not as accurate, still leads to appropriate clinical decision-making. Values falling within zones C, D, and E are defined to be outside of the acceptable range, potentially leading to incorrect treatment decisions or failure to detect hypoglycemia or hyperglycemia.

## 3. Results

Twenty-five patients were studied with a median age of 11.2 years (IQR: 9.2–14.0 years) and 16 patients (64%) were male. Twenty-four patients (96%) identified as white/Caucasian (Table 1). Of the 25 patients, 18 (72%) of them were found to have a known genetic mutation (PRSS1, SPINK1, CFTR, CTRC) as the etiology for acute recurrent or chronic pancreatitis. As discussed, any samples obtained after hydroxyurea initiation were excluded due to concern for medication interference [23]. The majority of the patients did not start hydroxyurea until after day 7, however a total of 12 patients had values excluded (total of 30 observations excluded) due to hydroxyurea administration. Two patients started the medication on day 4, two patients started it on day 5, five patients started it on day 6, and three patients started it on day 7. Including all time-matched samples prior to any hydroxyurea administration, there were a total of 225 time-matched samples during days 1–7 after TPIAT and 183 time-matched samples during days 2–7 after TPIAT. The overall MAD for days 1–7 was 16.2 mg/dL and the overall MARD for days 1–7 was 14.6%, with values of 19.8%, 15.2%, 14.0%, 12.1%, 11.4%, 13.2%, and 14.1% at days 1, 2, 3, 4, 5, 6 and 7, respectively (Table 2). Figure 1 demonstrates the variability of time-matched CGM and serum glucose values and shows the differences by day, including days 1–7.

When excluding day 1 and looking at days 2–7 after TPIAT, the MAD improved to 14.7 mg/dL, and the MARD improved to 13.4%. This dataset excludes day 1 to acknowledge this day as a “warm-up” day for the Dexcom G6 and to recognize that these values were likely to have more variability. Days 2–7 have a median difference of 9.0 (IQR, −4.0–18.0). Of these time-matched CGM values, 73% were within 20 mg/dL of serum glucoses, 58% were within 15 mg/dL, and 41% were within 10 mg/dL for days 2–7 post-TPIAT. The Dexcom G6 had a %20/20 accuracy of 78% and a %15/15 accuracy of 64% for days 2–7 post-TPIAT (Table 3). For serum glucose values, 95% were in goal range (70–140 mg/dL) and 5% were out-of-range (<70 or >140 md/dL) for days 2–7 post-TPIAT. For CGM values, 84% were in goal range (70–140 mg/dL) and 16% were out-of-range (<70 or >140 md/dL).

Time-matched values are displayed on a Clarke Error Grid plot in Figure 2. In this analysis, 77% of time-matched values were considered clinically accurate (zone A), and 100% of values were considered clinically acceptable (zones A and B). Therefore, this resulted in no difference in clinical decision-making.

## 4. Discussion

This single center analysis demonstrates that the Dexcom G6 CGM is a clinically acceptable and accurate tool for continuous glucose monitoring in the immediate postoperative period after TPIAT. Dexcom G6 CGM plays an important role in maintaining euglycemia in our TPIAT patients requiring strict glycemic control. This tool is useful in optimizing islet engraftment during this critical post-operative period.

Clarke Error Grid helps us understand how, despite reported values for %20/20 and %15/15 accuracy, the CGM values would compare to blood glucose values when it comes to clinical decision-making. We found that 100% of paired values from Clarke Error Grid analysis lie in zones A and B. This indicates that when comparing CGM values to serum glucose values, there would be no clinically significant difference in medical decision-making. Therefore, all CGM values obtained in this study, excluding day one, would result in clinically appropriate management.

The majority of the serum and CGM glucoses obtained in this study for days 2–7 were in a relatively euglycemic range with a serum glucose range (minimum-maximum) between 76 and 182 mg/dL and a CGM glucose range between 56 and 180 mg/dL. When discussing days 2–7 post-TPIAT, we propose that the Dexcom G6 may serve a useful role in maintaining the strict euglycemia needed for optimal islet engraftment in these patients, with almost all samples (95% by serum glucoses and 84% by CGM glucoses) remaining in goal glycemic range (70–140 mg/dL).

To further understand the Dexcom G6 accuracy in our post-TPIAT PICU patient population, we reported a %15/15 accuracy of 64% and %20/20 accuracy of 78%. Dexcom, however, cites a study that reports %15/15 and %20/20 of 83.3% and 93.9%, respectively [7]. Similarly, our overall MARD also differs from that reported by Dexcom. Our calculated MARD of 14.6% (which includes day 1) is greater than Dexcom’s calculated MARD of 9.0%. Furthermore, independent research into the Dexcom G6 reports similar MARDs to Dexcom with values of 10%, 10.3%, and 7.7% [8,9,10]. Compared to a similar study looking at the Dexcom G4 CGM, our calculated MARD is elevated compared with the previously reported MARD of 10.6% [6]. Despite the differences in reported MARD from Dexcom, as well as from prior G6 and G4 model comparison studies and despite our reported values of %20/20 and %15/15 accuracy, all of our time-matched samples during this study were considered either clinically accurate (77%) or clinically acceptable (100%), which, as mentioned previously, would result in clinically acceptable decision-making.

There was also a difference in the calculated MARD on day 1 of CGM use (19.8%), as compared to days 2–7 (13.4%). This seems to be partly accounted for by the aforementioned “warm-up” period that the Dexcom G6 expects with each new patient in order to achieve improved accuracy moving forward [24]. However, Dexcom manufacturers reported a MARD of 9.3% on day 1 which is only slightly elevated from their average MARD of 9.0% [7]. Our data include a very different patient population compared to what was studied in Dexcom’s above MARD, and perhaps our further variation in MARD is explained by the subcutaneous edema our patients likely develop on post-operative day one.

There were several limitations to this study. First, it is important to highlight the impact of sample size when using MARD for interpretation. Our sample size was limited to 25 patients and 183 paired samples in the immediate post-operative period where we expect to have vast fluid shifts and edema, possibly affecting Dexcom readings. As discussed, Dexcom, on the other hand, reported studies where they cite an improved MARD compared to our findings. This is likely due to a higher sample size of 66 patients with Types 1 or 2 diabetes, none of which were studied in an immediate post-operative period [7]. Since MARD becomes more precise as datasets increase, we believe our small sample size may have impacted our findings. Reiterer et al. has explained that MARD may not always be the best method for determining accuracy, especially when taking into account small sample sizes, as in our study [25]. Another consideration for limitations is that, to avoid potential interference from the surgical site and abdominal wall edema, the Dexcom G6 model was placed on the anterior thigh, which is not a manufacturer-recommended placement site at this time. This anterior thigh placement was necessary to avoid interference with appropriate post-operative wound care and to avoid potential CGM inaccuracy from insertion into extensive post-operative abdominal wall edema. Furthermore, this study also does not look at the accuracy of the Dexcom G6 in moderate to severe hyperglycemia and hypoglycemia which will require future studies to best illustrate the degree to which Dexcom can be used for clinical decision-making in these extremes. Finally, we did not have documented data on whether there were any failure rates of CGM technology which could have potentially affected our results if this existed in our study; this would be useful to document for future studies.

It is important to note that, as we look forward to using CGM technology in the post-op TPIAT population, the use of hydroxyurea in this population does not allow for CGM use alone for monitoring [22]. Tellez et al. has reported on the degree of discrepancy and duration of action of glucose sensor discrepancy after hydroxyurea dosing [23]. Dexcom has now made a statement instructing patients not to use their product during hydroxyurea administration [22]. At our institution, warning of hydroxyurea effect is discussed with hospital staff, patients, and their families. We continue to use the device to monitor trends 6–9 h after hydroxyurea administration, but all glycemic management decisions are based on glucometer data.

Use of Clarke Error Grid analysis proved important in this study. Despite the discrepancy in MARD percentage in these patients compared with Dexcom’s findings, the Clarke Error Grid analysis revealed that there would be no clinically significant difference in decision-making when comparing CGM to serum glucose values. Regardless, at this time our hospital protocol outlines the use of CGM to monitor trends but does not replace point-of-care or serum glucose testing while inpatient.

Although the goal of the Dexcom G6 CGM is to minimize the need for fingerstick glucoses (or the need for serum glucose comparisons), Dexcom also provides important data regarding glucose trends. This is especially important in the post-TPIAT patient population given the importance of maintaining euglycemia during the islet cell engraftment period. These data show that CGM monitoring paired with serum values without hydroxyurea therapy does not result in a clinically significant difference in decision-making despite the difference in MARD, %20/20 and %15/15 reported in this small study. As technology continues to advance, the Dexcom G6 has an opportunity to operate with an insulin infusion in a closed loop system. Eventually, we hope to use CGM technology in place of hourly serum or point-of-care glucose monitoring for this patient population. Nonetheless, the Dexcom G6 continues to aid this post-surgical patient population after leaving the PICU, transferring to the floor and after being discharged home from the hospital when fingerstick and serum glucoses are no longer checked as frequently.

## 5. Conclusions

The Dexcom G6 CGM may be an accurate tool producing clinically acceptable values to make reliable clinical decisions in the immediate post-TPIAT period before hydroxyurea therapy, as documented by percent in target and 100% of paired values in zones A and B of the Clarke Error Grid analysis. We found the MARD, %20/20 and %15/15 different than what Dexcom and other studies have reported which could potentially be due to our small sample size and to fluid shifts and edema in critically ill patients. Ultimately, we documented that 100% of our time-matched paired CGM and serum samples result in safe and correct decision-making for patient care. Safeguards with glucose monitoring must be in place especially during day 1 of Dexcom G6 use ensuring fingerstick or serum glucose monitoring prior to any adjustment in insulin therapy. As we continue to move toward advances, including a closed-loop insulin delivery system, it is important to understand the Dexcom G6 accuracy. Continued research is needed to fully understand and evaluate the accuracy of the G6 model in the ICU population post-TPIAT.

## Figures and Tables

**Figure 1 jcm-10-01893-f001:**
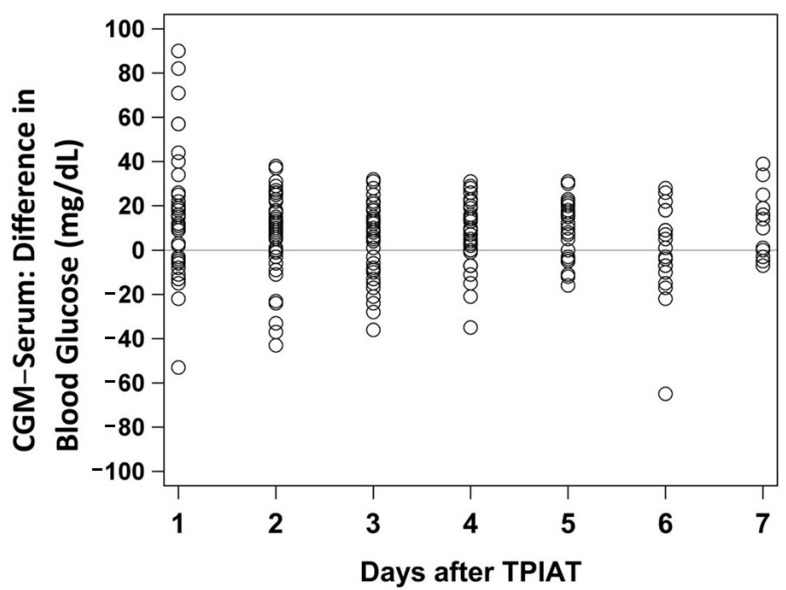
Serum glucose and CGM glucose reading differences for days 1–7. CGM = continuous glucose monitor. TPIAT = total pancreatectomy with islet autotransplantation.

**Figure 2 jcm-10-01893-f002:**
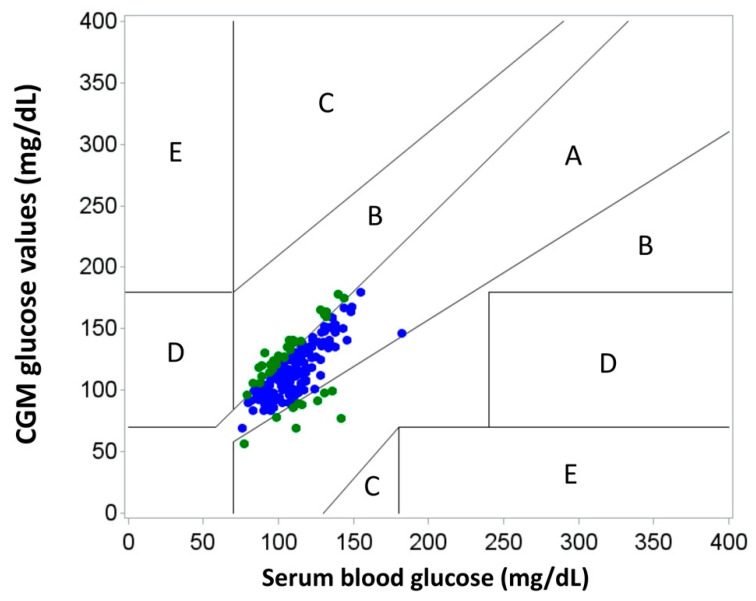
Clarke Error Grid analysis for days 2–7.

**Table 1 jcm-10-01893-t001:** TPIAT Patient Characteristics.

	TPIAT Patients
	*n* = 25
TPIAT age (years)	11.2 (9.2–14.0)
Sex (male)	16 (64%)
Race (White/Caucasian)	24 (96%)
Ethnicity (Non-Hispanic)	23/23 (100%)
Weight percentile	73.8 (60.7–90.9)
Height percentile	60.5 (32.0–71.3)
BMI percentile	81.9 (63.2–93.5)
Total islet equivalents/kg (IEQ/kg)	4518 (3154–7028)
Genetic testing positive	18 (72%)
PRSS1	10/22 (45%)
SPINK1	6/24 (25%)
CFTR	7/22 (32%)
CTRC	4/21 (19%)
More than 1 gene affected	7 (28%)
Exocrine insufficiency	8/24 (33%)

Data presented as median (25th–75th percentile) or *n* (%). TPIAT = total pancreatectomy with islet autotransplantation. BMI= body mass index.

**Table 2 jcm-10-01893-t002:** MARD values for Dexcom G6 overall and by day(s).

	MARD
Overall	14.6%
Day 1	19.8%
Day 2	15.2%
Day 3	14.0%
Day 4	12.1%
Day 5	11.4%
Day 6	13.2%
Day 7	14.1%

**Table 3 jcm-10-01893-t003:** Continuous glucose monitor (CGM) vs serum glucose data for days 2–7.

CGM vs. Serum	*n* = 183
Mean absolute difference (MAD) ± SD	14.7 ± 10.3
Mean absolute relative difference (MARD)	13.4%
Mean difference ± Standard Deviation	6.7 ± 16.7
Median difference (Interquartile Range)	9.0 (−4.0, 18.0)
Within ± 20 mg/dL	134 (73%)
Within ± 15 mg/dL	107 (58%)
Within ± 10 mg/dL	75 (41%)
%20/20	143 (78%)
%15/15	117 (64%)

## Data Availability

Data available on request due to privacy restrictions. The data presented in this study are available on request from the corresponding author.

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
