# Peer review of "Continuous Glucose Monitoring in the Intensive Care Unit Following Total Pancreatectomy with Islet Autotransplantation in Children: Establishing Accuracy of the Dexcom G6 Model"

_jcm, 2021, doi:10.3390/jcm10091893_

Round 1
Reviewer 1 Report
With the inclusion of 11 new patients (now 25), the main concern regarding this manuscrip has been addressed.
Author Response
Thank you for reviewing our manuscript again!
Reviewer 2 Report
The authors answered convincingly to all of my previously presented concerns.
Author Response
Thank you for reviewing our manuscript again!
This manuscript is a resubmission of an earlier submission. The following is a list of the peer review reports and author responses from that submission.
Round 1
Reviewer 1 Report
The authors evaluated the accuracy of the Dexcom G6 CGM in pediatric intensive care unit patients following total pancreatectomy with islet autotransplantation (TPIAT). They concluded that the Dexcom G6 CGM is an accurate tool producing clinically acceptable values to make reliable clinical decisions in the immediate post-TPIAT period.
I read the study with great interest. Unfortunately, I found significant methodological weaknesses. Also the study reveals some major flaws in the design and the presentation of the results.
Major objections:
- Primary and secondary outcomes of the study should be mentioned in the methodology.
- Inclusion / exclusion criteria should be mentioned in methodology.
- One of the biggest drawbacks of the present study is sample size. The sample of only 14 patients is too weak for significant conclusions to be made. Why the authors were in a hurry to publish the results? Why they did not include a larger number of patients in the study?
- Also, no sample size calculation was performed nor was the statistical power known.
- The authors stated that the Dexcom G6 model was placed on the anterior thigh, which is not a manufacturer-recommended placement site. This should be placed lateral to operative wound. Why on thigh? This should be source of the bias!
- An %20/20 accuracy of 78%, and a %15/15 accuracy of 60% should not be considered as a good result (if we compare to another studies with a larger sample size.
Minor objections:
- Citing references in the text [3,4,5,6,7,8] or [12,13,14,15] should be [3-8] or [12-15]. Please revise through the text
- Tables – All abbreviations used in Tables should be specified in the legend of each Table. The same is for Figures.
Reviewer 2 Report
The sample size used in the study (n=14) is considered insufficient. Furthermore, the variations found within the sample used make the results achieved unreliable. I recommend the extension of the retrospective cohort review period.
Reviewer 3 Report
Seger et al. were proving useful the monitoring of blood glucose by a Continuous Glucose Monitoring (CGM) device in children who underwent pancreatectomy, after otherwise unresolved chronic pancreatitis treatments.The article itself was well presented and demonstrated its point in a controlled setting.
Although it was not the focus of the paper, I believe it would be very useful for the reader to see a table indicating for the 14 patients, their age, weight, number of IEQ infused, and BG level in the first 7 days post-transplant. This will allow the reader to learn what to expect and when, in general, the transplanted beta cells start to kick in and indirectly timing their insulin secretion.
Also, considering the utility of CGM, the authors did not present enough data relative to the employment of blood sticks. I could see it being used, eventually, as an initial method of measurement, but replaced in situations in which the device is known to have issues. It would be useful to see when they were used too.
I don't know how much they would impact the article but there is some information, not included, that will help the reader to understand better the authors’ points. E.g.
Page 2 lines 49-51.
“… one of 2 FDA approved systems to replace finger stick glucose monitoring….”
It would be useful to know if the other system is indicated for children as well or if this is the only one. How is the accuracy compared to the other?
Page 2 lines 83-85.
“Extreme fluctuations of hypoglycemia are detrimental to islet engraftment. The primary objective is to establish Dexcom G6 in immediate postoperative period after TPIAT in PICU setting. The study monitors patients for 7 days in PICU.”
Is the 7-day period defined by when patients are discharged from PICU? When islet engraftment is done? Arbitrarily chosen? Do some/most/all patients become insulin independent within the 7-day period? Here again the suggested additional table could be useful.
Page 3 lines 106-107 and elsewhere
“…excluded from study after initiation of hydroxyurea due to potential interference of medication.”
It might be interesting to see these data in supplemental table to see how much of an effect there was.
More importantly, authors do not say how many time points were excluded in analysis, only that 131 samples remained after exclusion. Were samples excluded from all patients or only necessary in some? How many? Without knowing more about how common the use of hydroxyurea is in juvenile TPIAT patients and the extent of its use we don’t know how effective G6 will be with “typical” patient.
Page 4 line 143, table 2 MARD values days 1-7
Why are MARD values not recorded for days 3 or 6 and days 4-5 combined? Does this have to do with exclusion due to hydroxyurea?
Page 4, line 148-149.
“Day 1 values are excluded because they are variable and it is considered a “warm up day” for the G6.”
Does the previous G4 by same manufacturer have a similar warm up period? Is the other FDA approved system mentioned previously? It is important to know if this is a limitation in all CGM systems or just this device. If you are monitoring a patient in hospital for 7 days, excluding day 1 means the device cannot be relied on for 14% of the period of potential utility, and presumably day 1 would be important to recovery.
Page 6, lines 214-216.
“The site for testing was the thigh, which is not the recommended site by the manufacturer, but the surgeons justify it by keeping it clear of the surgical site.”
The author’s previous study of G4 showed that it was more effective than G6. Was the G4 test also in the thigh or a different location? Is there site-specific variability of readings that may make one site more accurate than another?
Also not addressed anywhere, is there any information on failure rate of the device for technical reasons like the sensor losing full contact or anything like that? Presumably that did not occur in the limited sample here but that could also be site dependent.
For accuracy, besides Bellin’s and Chinnakotla’s publications, the work of Cleveland’s group also should be quoted, e.g.
Bottino R, Bertera S, Grupillo M, Melvin PR, Humar A, Mazariegos G, Moser J, Walsh M, Fung J, Gelrud A, Slivka A, Soltys K, Wijkstrom, Trucco M: Isolation of human islets for autologous islet transplantation in children and adolescents with chronic pancreatitis. J Transplantation 2012:642787, 2012. PMCID: PMC3306977
Walsh RM, Aguilar Saavedra JR, Lentz G, Guerron AD, Schemen J, Stevens T, Trucco M, Bottino R, Hatipoglu B: Improved quality of life following total pancreatectomy and autoislet transplantation for chronic pancreatitis. J Gastrointest Surg 16:1469, 2012.
Johnston PC, Lin YK, Walsh RM, Stevens TK, Bottino R, Trucco M, Bena J, Faiman C, Hatipoglu BA: Factors associated with islet yield and insulin independence after total pancreatectomy and islet cell autotransplantation in patients with chronic pancreatitis utilizing off-site islet isolation: Cleveland Clinic experience. J Clin Enodocrinol Metabol 100(5):1765-70. doi: 10.1210/jc.2014-4298, 2015. PMID: 25781357